# Developing a Prognostic Gene Panel of Epithelial Ovarian Cancer Patients by a Machine Learning Model

**DOI:** 10.3390/cancers11020270

**Published:** 2019-02-25

**Authors:** Tzu-Pin Lu, Kuan-Ting Kuo, Ching-Hsuan Chen, Ming-Cheng Chang, Hsiu-Ping Lin, Yu-Hao Hu, Ying-Cheng Chiang, Wen-Fang Cheng, Chi-An Chen

**Affiliations:** 1Institute of Epidemiology and Preventive Medicine, Department of Public Health, National Taiwan University, Taipei 10055, Taiwan; tplu@ntu.edu.tw (T.-P.L.); ponponmiao@gmail.com (C.-H.C.); 2Department of Pathology, College of Medicine, National Taiwan University, Taipei 10002, Taiwan; pathologykimo@gmail.com; 3Department of Obstetrics and Gynecology, College of Medicine, National Taiwan University, Taipei 10041, Taiwan; mcchang@iner.gov.tw (M.-C.C.); wfc5166@gmail.com (H.-P.L.); jinmaw@gmail.com (Y.-H.H.); wenfangcheng@yahoo.com (W.-F.C.); chianchen@ntu.edu.tw (C.-A.C.); 4Institute of Nuclear Energy Research, Atomic Energy Council, Executive Yuan, Taoyuan 32546, Taiwan; 5Department of Obstetrics and Gynecology, National Taiwan University Hospital Yunlin branch, Yunlin 64041, Taiwan; 6Graduate Institute of Clinical Medicine, College of Medicine, National Taiwan University, Taipei 10002, Taiwan; 7Graduate Institute of Oncology, College of Medicine, National Taiwan University, Taipei 10002, Taiwan

**Keywords:** chemotherapy, microarray, ovarian cancer, predictive model, machine learning

## Abstract

Epithelial ovarian cancer patients usually relapse after primary management. We utilized the support vector machine algorithm to develop a model for the chemo-response using the Cancer Cell Line Encyclopedia (CCLE) and validated the model in The Cancer Genome Atlas (TCGA) and the GSE9891 dataset. Finally, we evaluated the feasibility of the model using ovarian cancer patients from our institute. The 10-gene predictive model demonstrated that the high response group had a longer recurrence-free survival (RFS) (log-rank test, *p* = 0.015 for TCGA, *p* = 0.013 for GSE9891 and *p* = 0.039 for NTUH) and overall survival (OS) (log-rank test, *p* = 0.002 for TCGA and *p* = 0.016 for NTUH). In a multivariate Cox hazard regression model, the predictive model (HR: 0.644, 95% CI: 0.436–0.952, *p* = 0.027) and residual tumor size < 1 cm (HR: 0.312, 95% CI: 0.170–0.573, *p* < 0.001) were significant factors for recurrence. The predictive model (HR: 0.511, 95% CI: 0.334–0.783, *p* = 0.002) and residual tumor size < 1 cm (HR: 0.252, 95% CI: 0.128–0.496, *p* < 0.001) were still significant factors for death. In conclusion, the patients of high response group stratified by the model had good response and favourable prognosis, whereas for the patients of medium to low response groups, introduction of other drugs or clinical trials might be beneficial.

## 1. Introduction

Ovarian carcinoma is a major cause of cancer death in women [1,2]. Due to the lack of initial symptoms and effective screening tools, most patients are diagnosed at an advanced stage with a 5-year survival of less than 50% [3,4]. The clinical prognostic factors include cancer stage, histological subtypes, tumor grade, the residual tumor size after debulking surgery and the response to chemotherapy. Despite good initial response, most ovarian cancer patients experience tumor recurrence and eventually are resistant to salvage treatments [3,5]. The serum CA-125 level is the current biomarker, but it is not ideal due to its limited specificity. Many potential biomarkers have been evaluated alone or in combination with CA-125, but the majority show disappointing results [6].

Precision medicine is the direction for cancer treatments, and the strategy of management depends on the distinct molecular features among these subtypes of ovarian carcinoma. However, the clinical benefits of targeted therapies are limited even though there are marked abnormalities in genetic or molecular pathways in ovarian cancer [7]. To date, the most promising agents are only anti-angiogenesis and poly ADP-ribose polymerase inhibitors. Bevacizumab, in combination with chemotherapy, demonstrates an improved progression-free survival, but a benefit on overall survival is only observed in high-risk patients [8,9]. Olaparib is approved for maintenance therapy in platinum-sensitive *BRCA*-mutated serous ovarian cancer patients [10]. The progression-free survival is significantly longer in platinum-sensitive recurrent ovarian cancer patients receiving niraparib, regardless of the *BRCA* mutation status [11]. In fact, the clinical dilemma is that these targeted drugs are expensive and benefit only a small subpopulation of ovarian cancer patients.

With the advancement in high-throughput genomic technology in recent years, many large-scale genomic and genetic studies have been performed to investigate cancer cell lines and patients [12,13]. For examples, the Cancer Cell Line Encyclopedia (CCLE) project analyzed more than 1000 cancer cell lines and reported their drug responses to more than 20 drugs and The Cancer Genome Atlas (TCGA) studied different kinds of genetic changes in real patients from more than 30 major cancer types. Due to the limitations from research ethnics and the difficulties in performing many clinical trials in real patients, these materials can serve as a good starting point to develop a machine learning model by using the cell lines first. Subsequently, the developed model can be validated in other large-scale studies, and such big data mining approach has been demonstrated its effectiveness in several studies [14,15].

Currently, cytotoxic chemotherapy still plays a key role in managing ovarian cancer. Determining how to predict the response of chemotherapy and identify which patients benefit from chemotherapy is important [16]. In this study, we initially developed a predictive model of chemo-response using CCLE and then selected for the optimal combination of predictors in TCGA and the GSE9891 dataset. Finally, we validated the model using clinical ovarian cancer tissue samples. We analysed the expression of the 10 predictive genes and correlated the clinical outcomes of the ovarian cancer patients to confirm the utility of the model.

## 2. Results

### 2.1. Development of A Predictive Model from Cancer Cell Line Encyclopaedia (CCLE)

After preprocessing, we utilized the Kruskal-Wallis test to identify the genes showing significant differences among the cell lines divided into three groups with an unequal drug efficacy in GSE36133 (Figure 1). The expression levels of 575 probes with a uniquely mapped gene name were significantly associated with the response (*p* < 0.01), and thus, they served as the original gene pool that might be possible predictive biomarkers. To identify a set of genes that could be used to develop a predictive model, we used the genetic algorithm (GA) shown in Appendix A to select the best combination of 10 genes from the 575 significant genes. A predictive model composed of the 10 selected genes was developed using the SVM algorithm, and its performance was evaluated based on the leave one out cross-validation strategy. Notably, the prediction accuracy gradually increased through the GA processes and became stable before 10 generations that were set as the last generation in the GA. The best combination of the 10 genes selected by the GA is summarized in Appendix A, and the corresponding SVM predictive model showed an accuracy of 100% for classifying the cell lines into three groups with different efficacies.

### 2.2. Evaluation of The Performance of The Developed Predictive Model

Even though the developed predictive model perfectly classified the cell lines into distinct groups corresponding to the efficacy, we still cannot exclude the possibility that the predictive model was identified purely based on a random chance. To address this issue, a permutation test was repeated 100,000 times to evaluate the probability of identifying a model with the same prediction accuracy. The details of the permutation test are described in the Methods section, and the results demonstrated that only 570 combinations out of the 100,000 trials showed 100% accuracy. Thus, the random chance of identifying a set of 10 genes showing 100% accuracy was only 0.0057. With such a low probability, the permutation test suggested that our proposed GA efficiently identified a set of genes with good prediction performances, and thus, the probability of identifying the developed predictive model is low.

### 2.3. Validation of The Predictive Model in Two Independent Datasets: TCGA and GSE9891

To validate the predictive model, two microarray datasets from patients with ovarian cancer were downloaded. One was from the TCGA, and another one was from the GEO (GSE9891). Because this study focused on the drug efficacy, only those patients receiving chemotherapy were used for the analyses. After preprocessing, the two microarray datasets were analysed using the developed SVM model. Therefore, each patient in the two datasets was classified into one of the three groups according to the predictive model. As shown in Appendix A, the high response group showed a better survival outcome, and no obvious differences were observed between the low response and medium response groups. To reduce the ambiguity, only the two groups with high and low responses are illustrated in Figure 2A,B. Significant differences in the RFS between the two groups were shown in the two datasets (log-rank test, *p* = 0.015 for TCGA and *p* = 0.013 for GSE9891), suggesting the effectiveness of the predictive model in predicting the chemo-response. Intriguingly, the proportion of patients showing a good response was 14.5% in TCGA and 19.3% in GSE9891, which concurred with the finding of less than 30% of a good response in the ovarian cancer patients receiving chemotherapy [17]. In addition to the RFS, we evaluated whether the developed SVM model was predictive for the OS. In general, the pattern of the three groups for the OS was similar to the RFS (Appendix A). Notably, a significant difference was observed in the TCGA dataset (log-rank test, *p* = 0.002, Figure 3A), whereas the OS was not significantly different in GSE9891 (Figure 3B). In addition, the proportion of patients showing a longer OS (>5 years) and RFS (>2 years) are summarized in Appendix A, and the proportion from the high response group was higher than that in the other two groups. The results showed that the SVM model we developed identified patients with a high response to chemotherapy, and a significant difference in the RFS was observed between the high and low response groups.

### 2.4. Validation of The Predictive Model by QRT-PCR in Ovarian Cancer Tissue of NTUH

In addition to the microarray datasets, we validated the prediction performance of the developed SVM model in another cohort of ovarian cancer patients. The study subjects were recruited from the NTUH, and the gene expression values of the 10 selected genes were measured using QRT-PCR. The clinical characteristics of this cohort are shown in Table 1. Similar to the previous validation in the microarrays, the patients were classified into three groups based on the developed SVM model after standardization. The survival curves for RFS and OS in the three groups from the NTUH patients were similar to the TCGA and GSE9891 datasets (Appendix A). As shown in Figure 2C and Figure 3C, significant differences were observed between the high and low response groups for both the RFS (log-rank test, *p* = 0.039) and the OS (log-rank test, *p* = 0.016), suggesting the effectiveness of the developed SVM model even if the experimental technology is different.

Lastly, we evaluated whether the developed SVM model was an independent predictor to known clinical factors (Table 2). A Cox hazard regression model was utilized to compare the predictive model with several clinical features, including residual tumor size < 1 cm, FIGO stage, tumor grade and histological subtypes. In univariate Cox regression, the predictive model (HR: 0.643, 95% CI: 0.415–0.998, *p* = 0.049), residual tumor size < 1 cm (HR: 0.273, 95% CI: 0.160–0.468, *p* < 0.001), advanced FIGO stage (HR: 7.954, 95% CI: 1.934–32.71, *p* = 0.004) and high tumor grade (HR: 2.289, 95% CI: 1.156–4.530, *p* = 0.018) were significant factors for recurrence. For death, the predictive model (HR: 0.559, 95% CI: 0.351–0.890, *p* = 0.014), residual tumor size < 1 cm (HR: 0.198, 95% CI: 0.107–0.365, *p* < 0.001), advanced FIGO stage (HR: 6.13, 95% CI: 1.489–25.24, *p* = 0.012) and clear cell carcinoma (HR: 0.284, 95% CI: 0.102–0.793, *p* = 0.016) were significant factors in univariate Cox regression. In multivariate Cox regression analysis, the predictive model (HR: 0.644, 95% CI: 0.436–0.952, *p* = 0.027) and residual tumor size < 1 cm (HR: 0.312, 95% CI: 0.170–0.573, *p* < 0.001) were significant factors for recurrence. For death, the predictive model (HR: 0.511, 95% CI: 0.334–0.783, *p* = 0.002) and residual tumor size < 1 cm (HR: 0.252, 95% CI: 0.128–0.496, *p* < 0.001) were still significant factors. As shown in Table 2, the developed SVM model was an independent and significant predictor after adjusting with these clinical factors. Therefore, the results indicated that the developed SVM model further improved the prediction of the patients’ chemo-response.

## 3. Discussion

Our model predicted the chemo-response in ovarian carcinoma patients. Several types of chemotherapy sensitivity and resistance assays (CSRAs) have been reported, such as the adenosine triphosphate assay, the human tumor cloning assay, the methylthiazolyldiphenyltetrazolium bromide assay, and the extreme drug resistance assay, as well as the drug-induced apoptosis assay, but the role of CSRAs remains controversial [18]. Other platforms, including proteomics [19], exosomes [20], next generation sequencing [21] and in vivo ovarian cancer patient-derived xenografts [22], also have weaknesses of inconsistent results, expensive costs and time-consuming processes. With the advancement of high-throughput technologies, researchers are now able to measure the gene expression profiles of one patient at a low cost. Therefore, considering the genetic features in one single individual may shed light on how to achieve the concept of providing precision medicine. However, it is difficult to develop a predictive model of one specific drug by investigating and manipulating the samples directly from real patients. Han et al. developed multiple gene predictive models for the platinum/paclitaxel response from the TCGA gene expression dataset [23]. Murakami et al. developed a multiple-gene scoring system for predicting a platinum or taxane response in ovarian cancer from the TCGA and GSE datasets [24]. However, both studies were not validated in ovarian cancer tissue samples by evaluating the clinical feasibility of the predictive models. To address this issue, we used cell lines as the identification set and validated the results in clinical samples [25].

In this study, we demonstrated the usefulness of the model in predicting the chemo-response, and survival benefits were observed in three independent datasets, including ovarian cancer tissue samples from our institute. Notably, relatively few samples from the NTUH cohort have been classified into “high” or “low” response groups. It can be mainly attributed to that we tried to classify the samples into three groups, which may result in some false positive classifications. However, as shown in Appendix A, the curves from the “medium” response group showed no significant differences to the curves from the “low” response group. This suggests that only the patients classified into “high” response group have definite benefits from receiving the drug treatment. For the other two groups, we should be more cautious while designing the treatment plan and selecting the appropriate drugs. In clinical practice, the cancer tissue of ovarian cancer patients could be tested by our 10 gene model before chemotherapy. For the patients of high response group, they have the best response to the paclitaxel-platinum chemotherapy which could be the standard treatment. However, in the patients of medium to low response groups, the response to the paclitaxel-platinum chemotherapy is not good enough, and early introduction or combination of other drugs should be considered.

The reason why we utilized a GA to select 10 possible predictors from 575 genes is its low computational complexity and good classification accuracy. Notably, the number of possible combinations of the original 575 genes is $C_{10}^{575}=1.01\times 10^{21}$, which cannot be analysed in a practical amount of time. The GA is a greedy approach that randomly selects and tests possible combinations through many generations. By doing so, the GA avoids calculating all the combinations and still can identify useful predictors in a reasonable amount of time [14]. Furthermore, in order to reduce the cost and simplify the experimental processes, we wished to minimize the number of predictors in the predictive model. However, the prediction accuracy was poor when the number of predictors was <10; thus, we defined it as 10.

Several studies demonstrated that the genes of our predictive model were involved in the growth, proliferation, and drug response of cancer cells [26,27,28,29,30,31]. We utilized the Ingenuity Pathway Analysis website to explore the possible interaction network among the 10 genes. As shown in Appendix A, five out of the 10 genes are summarized into one network centering on *TGFB1* and *E2F1* which are important regulators in the cell cycle. The results indicated that the genes of our predictive model have an important functional impact in ovarian cancer cells relating to chemo-response.

The study had some limitations. Notably, the standard regimen of chemotherapy for epithelial ovarian cancer is combination of platinum and paclitaxel. In this study, we only used the drug response efficacy of paclitaxel in the CCLE dataset to identify possible biomarkers, which might result in some biases and neglect the combination effects after treated with two different drugs. This was certain a potential limitation in our approach. However, the large-scale gene expression data along with the drug efficacy after treatment with these two drugs was not available currently. Therefore, it was not feasible to develop a prediction model by performing an integrated analysis of these two drugs simultaneously. In our identification set, the CCLE dataset, those studied ovarian cancer cell lines included both platinum-sensitive and platinum-resistant, such as ES-2, NIH:OVCAR-3, and SKOV-3. Therefore, those identified genes reflected the effect which was not solely from paclitaxel. Validation of this prediction model in three datasets in real clinical patients also showed a good performance, suggesting that although the approach in this study was not a perfect one, the resulted prediction model still had a good utility in daily clinical practice. The other limitation was the different definition of RFS in TCGA, GSE9891 and our set. RFS was calculated from the completion of adjuvant chemotherapy and we used the definition in our own NTUH dataset. However, in the data from the public domains including GSE9891 and the TCGA dataset, we used those provided RFS information along with their microarray data. However, this might result in some potential biases from different definitions of the RFS. Lastly, it was a limitation that the NTUH cohort did not include the tissue samples of the non-ovarian cancer group, which may help to provide more detailed information of our gene model about the chemo-response of ovarian cancer.

## 4. Materials and Methods

### 4.1. Identification of The Genes Associated with The Drug Response

The protocol used to identify the probes associated with the drug efficacy and develop the predictive model is illustrated in Figure 1. The details about selecting predictive biomarkers from GSE36133 (Appendix A) were described in the Appendix A. Briefly, after pre-processing, we analysed the 25 ovarian cancer cell lines (Appendix A) to identify the genes associated with the efficacy of paclitaxel treatment. The 25 ovarian cancer cell lines were classified into three groups based on their sensitivity, which was the activity area provided by the GSE36133 dataset. Furthermore, a quantile normalization algorithm was performed and a Kruskal-Wallis test was utilized to identify the probes showing significantly different expression levels in the three groups (*p* < 0.01). Only the probes mapped to a unique gene symbol remained for further analyses. When multiple probes were annotated to the same gene, their coefficients of variation (CVs) were calculated, and only the probe possessing the largest CV was kept. In order to reduce systematic biases resulted from different technological platforms, the expression value of each gene was normalized to Z-value based on its mean and standard deviation, respectively.

### 4.2. Development of a Predictive Model Using a Genetic Algorithm

Among the probes showing a significant association with the response of paclitaxel, a genetic algorithm (GA) was designed to select the best combination of 10 probes to classify the 25 cell lines into the three groups, corresponding to the drug efficacy (Appendix A). The detailed procedures in the GA were described in the Appendix A. In general, the GA algorithm mimics the concept of the “survival of the fittest,” which indicates that a combination showing a better prediction performance has a higher probability of being selected in the next generation. Therefore, the model showing the highest accuracy for predicting the paclitaxel response was developed in the last generation. We also performed a permutation test to evaluate the random chance of identifying 10 probes with the same prediction accuracy.

The definitions of response groups were based on the activity area of the drug efficacy provided in the CCLE dataset. We sorted the values ascendingly and divided them into three groups accordingly. That is after being sorted, the first nine cell lines with lower activity area values from the 25 ovarian cancer cell lines were classified into “low” response group (i.e., activity area less than 4 in Appendix A). Subsequently, the eight cell lines with medium activity values were classified into “medium” response group and the last eight cell lines with higher activity values were classified into “high” response group (i.e., activity area higher than 5.29 in Appendix A).

### 4.3. Validation of the Predictive Model in Two Independent Datasets

In addition to the internal validations using permutation and cross-validation in the CCLE dataset, two ovarian cancer microarray datasets from TCGA [13] and GSE9891 [32], composed of real clinical samples, were analysed (Appendix A). To focus on the chemo-response, only those patients who received the drug treatment and survived longer than 30 days were analysed. All the analysis procedures followed the same steps previously described, and all the patients were classified into distinct groups using the predictive model. The gene expression data from the microarray datasets were normalized to the standard normal distribution, and the drug response of one patient was predicted by the developed SVM model that was composed of the 10 selected probes. For the patient groups with different drug efficacies, the log-rank test was utilized to evaluate whether significant differences existed in the overall survival (OS) and/or recurrence-free survival (RFS).

### 4.4. Validation of the Predictive Model by a Quantitative Real-Time Polymerase Chain Reaction (QRT-PCR) Using Ovarian Cancer Tissue

Lastly, we validated the predictive model using the QRT-PCR method in patients recruited from the National Taiwan University Hospital (NTUH). The study protocol was approved by the National Taiwan University Hospital Research Ethics Committee. Informed consents were obtained and the methods were performed in accordance with the guidelines and regulations. From January 2012 to March 2014, 84 women diagnosed with ovarian carcinoma who received debulking surgery and adjuvant chemotherapy were enrolled. Part of cancerous tissue specimen collected in debulking surgery was immediately frozen in liquid nitrogen and stored at −70 °C. The remaining tissue specimens were sent for frozen section and pathology examinations to confirm the diagnosis and ensure sufficient tumor tissue in the specimens collected for the following experiments. The medical records of the patients were reviewed until June 2017 to obtain clinical data, including the age, cancer stage, surgical findings during debulking, treatment courses, recurrence and survival. Residual tumor size was recorded as <1 cm or ≥1 cm after debulking surgery. The tumor grading was based on the International Union Against Cancer criteria, and the staging was based on the criteria of the International Federation of Gynecology and Obstetrics (FIGO) [33]. The patients received regular follow-ups every 3 months after the primary treatment. Abnormal results from imaging studies (including computerized tomography or magnetic resonance imaging), elevated CA-125 (more than 2 times the upper normal limit) for two consecutive tests in 2-week intervals, or biopsy-proven disease were defined as recurrence. The RFS was calculated as the period from the date of chemotherapy completion to the date of confirmed recurrence, disease progression, or the last follow-up. The OS was calculated as the period from the surgery to the date of death associated with the disease or the date of the last follow-up. To evaluate the required the sample size for the validation, we utilized the estimated hazard ratio from the TCGA data (HR = 0.52). Therefore, using the parameters (type 1 error: 0.05, power: 70%, overall probability of event: close to 1, and the proportion of samples in “low” risk group: 0.776, the estimated sample size is 84.

Total RNA of the cancerous tissue was isolated with TRIzol reagent (Invitrogen Corporation, Carlsbad, CA, USA) according to the manufacturer’s instructions. The samples were subsequently passed over a Qiagen RNeasy column (Qiagen, Valencia, CA, USA) to remove the small fragments that affect the RT reaction and hybridization quality. After RNA recovery, double-stranded cDNA was synthesized by a chimeric oligonucleotide with an oligo-dT and a T7 RNA polymerase promoter at a concentration of 100 pmol/µL. The protocol for the quantitative real-time polymerase chain reaction is briefly described. First-strand cDNA was synthesized with a RevertAid first strand cDNA synthesis kit (Fermentas, Burlington, ON, Canada; Vilnius, Lithuania). Quantitative PCR was performed using the LightCycler Real-Time detection system (Roche Diagnostics, Mannheim, Germany). The relative abundance of the mRNA level was calculated by using the comparative method with glyceraldehyde-3-phosphate dehydrogenase (*GAPDH*) as the internal control. The quantitative PCR primers for the TaqMan probes are listed in Appendix A. The detection of *GAPDH* was carried out by the LightCycler h-*GAPDH* housekeeping gene set (Roche Applied Science, Indianapolis, IN, USA) for 50 cycles of 10 s at 95 °C, 15 s at 55 °C, and 15 s at 72 °C.

The expression levels of the 10 probes were normalized to the standard normal distribution. Following the prediction procedures previously described, each patient was classified into a different group based on the SVM predictive model. The differences in the OS and RFS in the predicted groups were compared using the log-rank test, and a Cox hazard regression model was utilized to evaluate the prediction performance.

## 5. Conclusions

The epithelial ovarian cancer patients of high response group stratified by our 10 gene model had good response to the paclitaxel-platinum chemotherapy which could be the standard treatment. However, for the patients of medium to low response groups, introduction of other drugs or clinical trials might be beneficial.

## Figures and Tables

**Figure 1 cancers-11-00270-f001:**
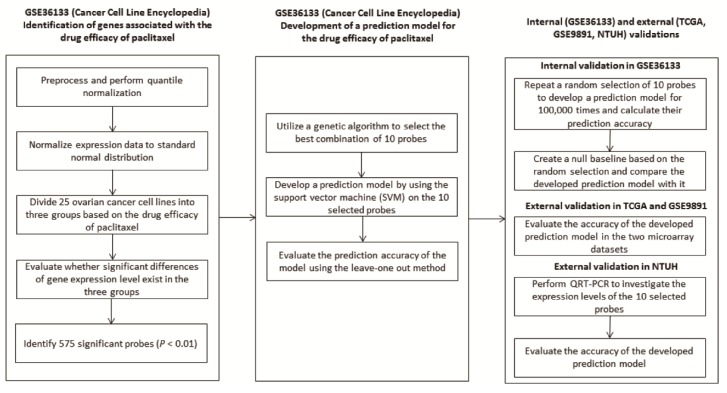
The overall protocol utilized in this study to develop the predictive model.

**Figure 2 cancers-11-00270-f002:**
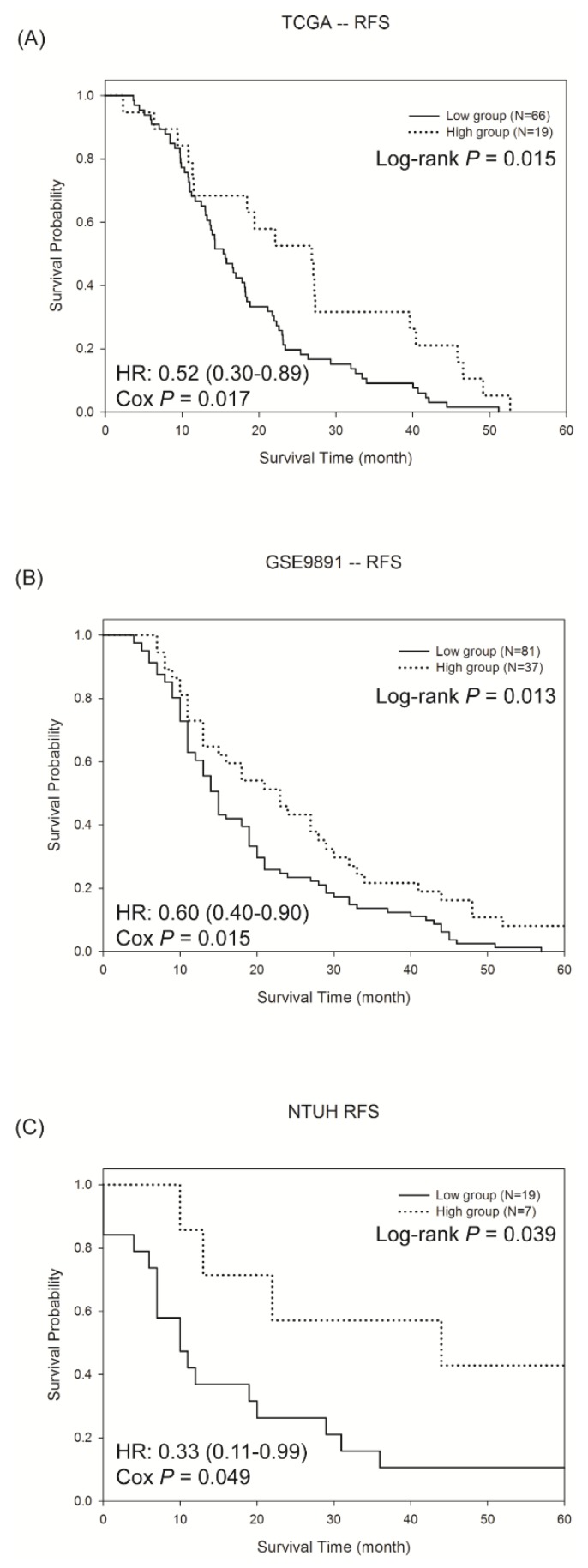
The Kaplan-Meier survival curves of the recurrence free survival (RFS) in the high and low response groups in the (**A**) TCGA dataset, (**B**) GSE9891 dataset and (**C**) NTUH patients. Patients of high response group had longer RFS than low response group in the three cohorts. The hazard ratio (HR) was estimated by the Cox hazard regression model.

**Figure 3 cancers-11-00270-f003:**
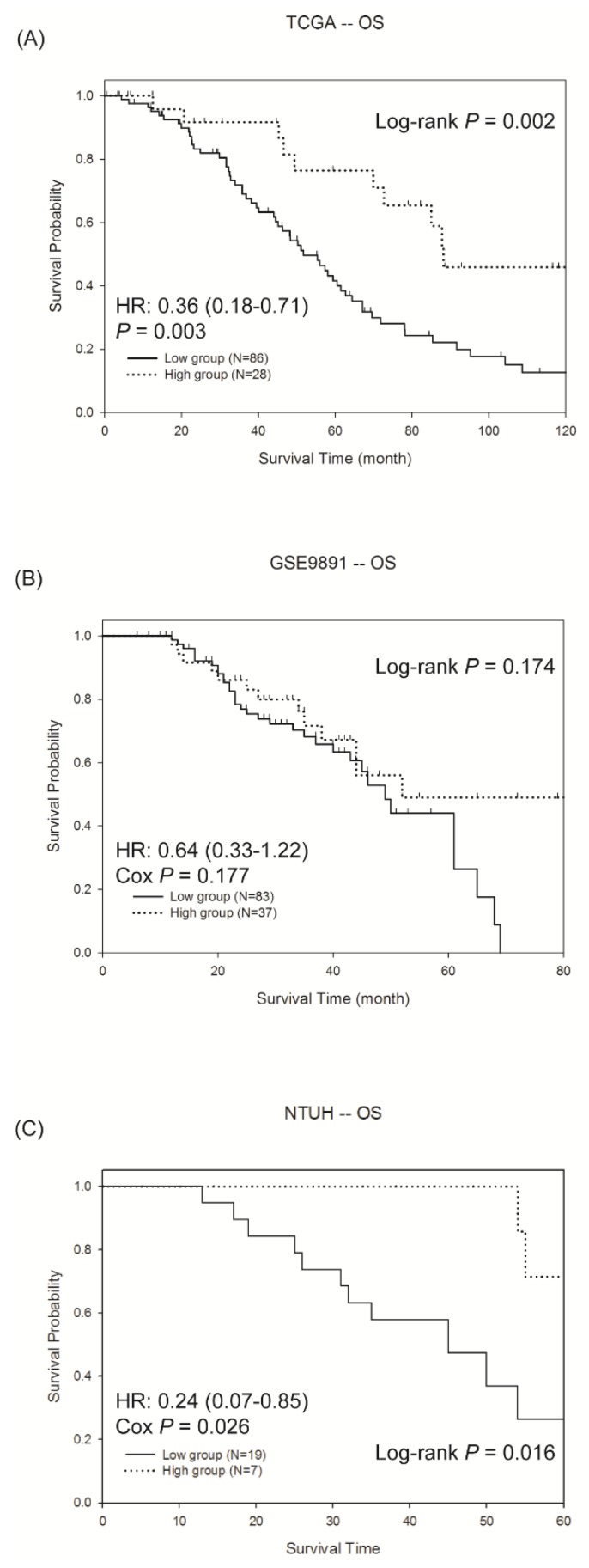
The Kaplan-Meier survival curves of the overall survival (OS) in the high and low response groups in the (**A**) TCGA dataset, (**B**) GSE9891 dataset and (**C**) NTUH patients. Patients of high response group had longer OS than low response group in the three cohorts. The hazard ratio (HR) was estimated by the Cox hazard regression model.

**Table 1 cancers-11-00270-t001:** Clinical characteristics of the epithelial ovarian cancer patients at NTUH.

Feature	Number (Proportion)
Total patients	84
Age (years)	54.94 ± 11.25
Histological subtype	
Serous	59 (0.7)
Endometrioid	12 (0.14)
Clear cell	13 (0.16)
Tumor grade	
1	4 (0.05)
2	10 (0.12)
3	57 (0.68)
Not available	13 (0.15)
FIGO_stage	
Early (I + II)	11 (0.13)
Advanced (III + IV)	73 (0.87)
Debulking surgery	
Residual tumor size < 1 cm	42 (0.5)
Residual tumor size ≥ 1 cm	42 (0.5)

**Table 2 cancers-11-00270-t002:** Cox regression model for the clinical features and predictive model in recurrence and death of NTUH patients.

Feature	Numbers	Recurrence				Death			
		Univariate		Multivariate		Univariate		Multivariate	
		HR (95% CI)	*p* Value	HR (95% CI)	*p* Value	HR (95% CI)	*p* Value	HR (95% CI)	*p* Value
**Predictive model**									
Low	19	1.00		1.00		1.00		1.00	
Medium + High	65	0.643 (0.415–0.998)	0.049	0.644 (0.436–0.952)	0.027	0.559 (0.351–0.890)	0.014	0.511 (0.334–0.783)	0.002
**Residual tumor size < 1 cm**									
No	42	1.00		1.00		1.00		1.00	
Yes	42	0.273 (0.160–0.468)	<0.001	0.312 (0.170–0.573)	<0.001	0.198 (0.107–0.365)	<0.001	0.252 (0.128–0.496)	<0.001
**FIGO stage**									
Early (I + II)	11	1.00		1.00		1.00		1.00	
Advanced (III + IV)	73	7.954 (1.934–32.71)	0.004	2.149 (0.257–17.93)	0.480	6.13 (1.489–25.24)	0.012	1.732 (0.201–14.90)	0.617
**Tumor grade**									
1	4	1.00		1.00		1.00		1.00	
2 + 3	67	2.289 (1.156–4.530)	0.018	2.125 (0.992–4.552)	0.053	1.756 (0.953–3.234)	0.071	1.533 (0.761–3.09)	0.232
**Histological subtype**									
Serous	59	1.00		1.00		1.00		1.00	
Endometrioid	12	0.454 (0.193–1.067)	0.070	1.023 (0.422–2.475)	0.960	0.562 (0.239–1.323)	0.187	1.077 (0.443–2.618)	0.870
Clear cell	13	0.531 (0.239–1.180)	0.120	NA	NA	0.284 (0.102–0.793)	0.016	NA	NA

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
