# Peer review of "Developing a Prognostic Gene Panel of Epithelial Ovarian Cancer Patients by a Machine Learning Model"

_cancers, 2019, doi:10.3390/cancers11020270_

Reviewer 1 Report

"Developing Prognostic Gene Panel of Epithelial Ovarian Cancer Patients by a Machine Learning Model",
by Tzu-Pin Lu et al

Overall, this is a technically solid paper with significant findings and addressing an important clinical disease which has no adequate predictive biomarkers
The methods used by the authors are standard and have be validated in other applications.

The authors were creative in their use of public datasets, genetic algorithms, machine learning and survival analysis methods.  

The data mining, biostatistics, and machine learning methods they used are all relatively well-established and uncontroversial.  Perhaps it is not strictly speaking "AI/ML", but that definition is a moving target. I suggest an examination of the title so it is not just a "hype".

Their description of the provenance and handling of the NTUH samples was thorough and clear but it is focused only on a subset of patients. Would been more clinically significant or useful if they had included a non -ovarian cancer group. 

I found the following elements of their investigation worthy of note:
1.  Their use of a genetic algorithm combined with SVM for selecting the best combination of 10 genes out of 575 candidates.
2.  The "permutation test" which they conducted to establish the (very  low) likelihood that this selection was accidentally good.

3.  The use of two independent public datasets, as well as the NTUH.

4.  The careful comparison of the results from those three datasets.

5.   Any clinical samples been tested instead of cell line. 

Although the survival analysis results are significant, what is needed to justify the next steps, whatever those may be, toward creating a practical diagnostic tool for physicians?  Please indicate how the method can be expanded to the clinic.

A few minor comments: 

 - In Figure 2 there are "587 significant probes associated...." but in Figure 1 only 575 probes were identified. 

-  How are the recurrence criteria defined? data bases vs NTUH? 

 - I can't find the 10 genes listed in this paper in other ML-based papers in the literature. Are there any references?   

Author Response

Response to Reviewer 1 Comments

Point 1: The data mining, biostatistics, and machine learning methods they used are all relatively well-established and uncontroversial.  Perhaps it is not strictly speaking "AI/ML", but that definition is a moving target. I suggest an examination of the title so it is not just a "hype".

Response 1: We appreciate the suggestion and re-examined the title of this manuscript and we believed it clearly reflected the methodology that we utilized a machine learning algorithm to develop the prediction model. So, we prefer to use the same title of this manuscript.

Point 2: Their description of the provenance and handling of the NTUH samples was thorough and clear but it is focused only on a subset of patients. Would been more clinically significant or useful if they had included a non -ovarian cancer group. 

Response 2: We appreciate the suggestion. We totally agree with the point that a non-ovarian cancer group may help to improve the prediction model. However, it was a limitation that the NTUH cohort did not include the tissue samples of the non-ovarian cancer group, which may help to provide more detailed information of our gene model about the chemo-response of ovarian cancer. We added last sentence to discussion to describe this limitation. (Please see Page 10 Line 239-241)

Point 3: Although the survival analysis results are significant, what is needed to justify the next steps, whatever those may be, toward creating a practical diagnostic tool for physicians?  Please indicate how the method can be expanded to the clinic.

Response 3: We appreciate the suggestion and added the following sentences to the manuscript. “In clinical practice, the cancer tissue of ovarian cancer patients could be tested by our 10 gene model before chemotherapy. For the patients of high response group, they have the best response to the paclitaxel-platinum chemotherapy which could be the standard treatment. However, in the patients of medium to low response groups, the response to the paclitaxel-platinum chemotherapy is not good enough, and early introduction or combination of other drugs should be considered.” (Please see Page 9 Line 202-206)

Point 4: In Figure 2 there are "587 significant probes associated...." but in Figure 1 only 575 probes were identified. 

Response 4: We apologized and corrected for the typo. The number should be 575 in original Figure 2. We had moved the original Figure 2 to supplementary data in the revised manuscript. (Please see Figure S1)

Point 5: How are the recurrence criteria defined? data bases vs NTUH? 

Response 5: We appreciate the suggestion. The recurrence for all microarray datasets retrieved in the public domain followed their definitions that were described in their original manuscript. That is we directly used the recurrence dates/durations along with the datasets and no modification was applied. For our own data, the NTUH recurrence criteria were described in “Materials and Methods”. The patients received regular follow-ups every 3 months after the primary treatment. Abnormal results from imaging studies (including computerized tomography or magnetic resonance imaging), elevated CA-125 (more than 2 times the upper normal limit) for two consecutive tests in 2-week intervals, or biopsy-proven disease were defined as recurrence. (Please see Page 11 Line 300-304)

Point 6: I can't find the 10 genes listed in this paper in other ML-based papers in the literature. Are there any references?

Response 6: We appreciate the suggestion and did a literature survey about the 10 genes related to the machine learning field and found these two papers. A previous study also used the CCLE dataset to predict the drug response and found DPEP2 as an important predictor of the response rate of the drug TKI258 [1]. Another machine learning study reported the methylation level of LRRC32 may be associated with tobacco smoking in a veteran HIV-positive population [2]. However, the second study was beyond the scope of this manuscript, and thus only the first study was added into the revised manuscript. (Please see Page 13 Line 404-405)

References

1.         Fang, Y.; Qin, Y.; Zhang, N.; Wang, J.; Wang, H.; Zheng, X. DISIS: prediction of drug response through an iterative sure independence screening. PloS one 2015, 10, e0120408, doi:10.1371/journal.pone.0120408.

2.         Zhang, X.; Hu, Y.; Aouizerat, B.E.; Peng, G.; Marconi, V.C.; Corley, M.J.; Hulgan, T.; Bryant, K.J.; Zhao, H.; Krystal, J.H., et al. Machine learning selected smoking-associated DNA methylation signatures that predict HIV prognosis and mortality. Clinical epigenetics 2018, 10, 155, doi:10.1186/s13148-018-0591-z.

Reviewer 2 Report

In this study, the author developed a predictive model for ovarian cancer starting from the CCLE dataset, with the specific focus on drug efficacy. They then evaluated their model by employing more data from TCGA and GEO. A ten-gene panel was reported with significant prediction accuracy. Overall, it's a well organized, written, and demonstrated study paper using the machine learning approaches for the pursuit of clinical insights of ovarian cancer. It's acceptable for publication if my following concerns were addressed appropriately.

Major:

1. Because the primary model was derived from CCLE, of which the data come from cell lines that are much more homogeneous than real patient samples. How to evaluate the effect of the heterogeneous of real patient sample to the model performance. The author did such types analyses shown in Table 2, however, did not clarify how other clinic characteristics affected the model performance because many of these clinic characteristics also have significant relevance with the recurrence and death. I would like to see how the authors could adjust the model performance after removal of the effects of all other clinic features listed in Table 2.

Minor:

The full name of 'GA' should be given at the first place where appears on page 2.

Figure 2 is not necessary to be included in the main text. I prefer to putting it in supplementary figures.

Author Response

Response to Reviewer 2 Comments

Point 1: Because the primary model was derived from CCLE, of which the data come from cell lines that are much more homogeneous than real patient samples. How to evaluate the effect of the heterogeneous of real patient sample to the model performance. The author did such types analyses shown in Table 2, however, did not clarify how other clinic characteristics affected the model performance because many of these clinic characteristics also have significant relevance with the recurrence and death. I would like to see how the authors could adjust the model performance after removal of the effects of all other clinic features listed in Table 2.

Response 1: We appreciate the suggestion and the comparisons of the prediction model with and without other clinical features including the subtypes and other important factors, such as residual tumor size, FIGO stage, and tumor grade, were reported in Table 2. We did both univariate and multivariate Cox regression models to evaluate the prediction model and other clinical features for RFS and OS, respectively. Notably, the p-value and the hazard ratio of the prediction model improved after considering other clinical features. That is the p-value and the hazard ratio for RFS increased from HR: 0.643 (p-value: 0.049) in the univariate model to HR: 0.644 (p-value: 0.027) in the multivariate model which considered all clinical features and our gene panel simultaneously. Similarly, the p-value and the hazard ratio for OS increased from HR: 0.559 (p-value: 0.014) in the univariate model to HR: 0.511 (p-value: 0.002) in the multivariate model which considered all clinical features and our gene panel simultaneously. Therefore, the results demonstrate that our gene prediction model is a significant and independent predictor even if other clinical features were considered together.

Point 2: The full name of 'GA' should be given at the first place where appears on page 2.

Response 2: We appreciate the suggestion and added the full name of “GA” in the revised manuscript. (Please see Page 2 Line 81)

Point 3: Figure 2 is not necessary to be included in the main text. I prefer to putting it in supplementary figures.

Response 3: We appreciate the suggestion and moved the original Figure 2 to supplementary data. (Please see Figure S1)

Round  2

Reviewer 2 Report

I am satisfied with the authors' response and recommend an acceptance for publication.